# Integrating the Soil Microbiota and Metabolome Reveals the Mechanism through Which Controlled Release Fertilizer Affects Sugarcane Growth

**DOI:** 10.3390/ijms241814086

**Published:** 2023-09-14

**Authors:** Zhaonian Yuan, Qiang Liu, Lifang Mo, Ziqin Pang, Chaohua Hu

**Affiliations:** 1National Engineering Research Center for Sugarcane, Fujian Agriculture and Forestry University, Fuzhou 350002, China; 18888425925@163.com (L.M.); ziqintea@126.com (Z.P.); chhu@fafu.edu.cn (C.H.); 2Province and Ministry Co-Sponsored Collaborative Innovation Center of Sugar Industry, Nanning 530000, China; 3College of Natural Resources and Environment, Northwest A&F University, Yangling 712100, China; fjnldxlq1190101005@163.com

**Keywords:** sugarcane, controlled release fertilizer, rhizosphere microorganisms, metabolites, application rate, sugar industry

## Abstract

Root−soil underground interactions mediated by soil microorganisms and metabolites are crucial for fertilizer utilization efficiency and crop growth regulation. This study employed a combined approach of soil microbial community profiling and non-targeted metabolomics to investigate the patterns of root-associated microbial aggregation and the mechanisms associated with metabolites under varying controlled-release fertilizer (CRF) application rates. The experimental treatments included five field application rates of CRF (D1: 675 kg/ha; D15: 1012.5 kg/ha; D2: 1350 kg/ha; D25: 1687.5 kg/ha; and D3: 2025 kg/ha) along with traditional fertilizer as a control (CK: 1687.5 kg/ha). The results indicated that the growth of sugarcane in the field was significantly influenced by the CRF application rate (*p* < 0.05). Compared with CK, the optimal field application of CRF was observed at D25, resulting in a 16.3% to 53.6% increase in sugarcane yield. Under the condition of reducing fertilizer application by 20%, D2 showed a 13.3% increase in stem yield and a 6.7% increase in sugar production. The bacterial ACE index exhibited significant differences between D25 and D1, while the Chao1 index showed significance among the D25, D1, and CK treatments. The dominant bacterial phyla in sugarcane rhizosphere aggregation included Proteobacteria, Actinobacteriota, and Acidobacteriota. Fungal phyla comprised Rozellomycota, Basidiomycota, and Ascomycota. The annotated metabolic pathways encompassed biosynthesis of secondary metabolites, carbohydrate metabolism, and lipid metabolism. Differential analysis and random forest selection identified distinctive biomarkers including *Leotiomycetes*, *Cercospora*, *Anaeromyxobacter*, isoleucyl-proline, and methylmalonic acid. Redundancy analysis unveiled soil pH, soil organic carbon, and available nitrogen as the primary drivers of microbial communities, while the metabolic profiles were notably influenced by the available potassium and phosphorus. The correlation heatmaps illustrated potential microbial−metabolite regulatory mechanisms under CRF application conditions. These findings underscore the significant potential of CRF in sugarcane field production, laying a theoretical foundation for sustainable development in the sugarcane industry.

## 1. Introduction

As the world’s most important sugar crop, sugarcane (*Saccharum officinarum* L.) has received much attention for its cultivation and production [1]. China is ranked third in the world for sugarcane cultivation area, but due to the long growing season of sugarcane, the cultivation process requires a large and continuous supply of fertilizer, which leads to an increase in cultivation costs for sugarcane farmers, poor economic benefits, and a consequent reduction in the area under cultivation [2]. In addition, the application of large quantities of fertilizer leads to the loss of effective soil nutrients, acidification, and hardening of the soil, which exacerbates the ecological burden and at the same time seriously threatens the development of the sugar industry [3]. Therefore, a rational and scientific fertilizer application program is of great importance for solving the current difficulties of sugarcane cultivation.

Controlled release fertilizer (CRF) is a new type of fertilizer that can release nutrients slowly and continuously to support plant growth [4]. Compared with traditional fertilizers, CRF releases nutrients at a stable rate over a period of time through special packaging materials, thus meeting the needs of plant growth and reducing nutrient wastage and environmental pollution [5,6]. In crop production, the application of CRF has been widely promoted. It is widely used in the production of vegetables, fruits, cereals, cash crops, and other types of crops [7,8,9]. Studies have shown that through the rational selection and application of CRF, the nutrient utilization rate of crops can be improved, fertilizer wastage and pollution can be reduced, and the yield and quality of crops can also be improved [10]. Thus, it is important to elucidate the applicability and mechanism of action of CRF in sugarcane and to investigate the synergistic effects of sugarcane soil metabolites and microorganisms under CRF application conditions, which will contribute to the future development of the sugar industry.

Soil microorganisms and metabolite−plant interactions have an important influence on the acquisition of high-yielding plant phenotypes [11]. Functional soil microorganisms promote plant growth through the production of plant growth hormones (e.g., glutathione and gibberellin), and which regulate root morphology and plant nutrient uptake, thereby directly increasing the plant growth rate and yield [12,13]. The production of antioxidants and stress proteins by the rhizosphere microbial community can help to increase plant stress tolerance [14], thus providing strong support for plants to cope with environmental stresses such as drought, salinity, pests, and diseases, and at the same time contribute to the establishment of rhizosphere homeostasis [15]. In addition, microbiota release nutrients such as nitrogen, phosphorus, and potassium by decomposing organic matter and dissolving minerals, thereby increasing the proportion of nutrients available to the soil [16]. Specific beneficial functional microorganisms (e.g., rhizobacteria and arbuscular mycorrhizal fungi) can form unique symbiotic relationships with the plant root system to enhance nutrient uptake and utilization by the plant [17]. Mechanisms such as resource competition between microorganisms and the induction of plant immunity to inhibit the growth and spread of plant pathogens act as biocontrol agents to protect plant health [18]. Metabolite exchange between microorganisms can also influence the synthesis of the root chemicals in plants [19]. For example, functional microorganisms promote plant synthesis of secondary metabolites such as antioxidants, antimicrobials, and medicinal compounds [20,21]. Sugarcane, as a photosynthetically vigorous high-carbon species, produces abundant secondary metabolites during its growth and development, including sugars, ketoacids, amino acids, etc. [22]. These metabolites are not only important indicators of sugarcane growth and development, but are also associated with plant adaptation to adverse conditions and rhizosphere microbial interactions [23]. Advances in research on soil microbes and metabolite−plant interactions have revealed the complex interrelationships and mechanisms of action between microbes and plants [24,25,26]. These studies provide new ideas and strategies for agricultural production and environmental stewardship that can help to improve plant growth and resistance, reduce the use of chemical pesticides, and promote sustainable agricultural development.

In conclusion, this study investigates the effect of CRF application on sugarcane growth using high-throughput soil sequencing and non-target metabolome analysis to elucidate the mechanism responsible for regulating CRF application in sugarcane. The study addresses three questions, which are detailed and clarified. (i) Determination of the optimal CRF application, as well as the composition and diversity of rhizosphere microbial communities during sugarcane cultivation. (ii) Screening of characteristic microorganisms and metabolites under varying CRF application rates. (iii) Potential correlations between rhizosphere microbial and metabolites, and drivers of major nutrient factors for sugarcane rhizosphere soil compounds and microbial communities. Our study presents a sound molecular theoretical foundation for the extensive implementation of CRF in sugarcane cultivation.

## 2. Results

### 2.1. Response of Agronomic Traits of Sugarcane to Different CRF Application Rates

The growth of sugarcane in the field was significantly affected (*p* < 0.05) by the application rate of CRF. Under D25 and D3 conditions, the sugarcane plant height and stem diameter indexes peaked, respectively. Under moderate D25 conditions, the sucrose content reached its peak. The maximum yield of cane stems in the field corresponded to the highest application rate of CRF, but the sugar yield, which is the ultimate objective, corresponded to the CRF application under D25 condition (Table 1). The different application rates of CRF reflected the advantages and disadvantages of field cane stem yield and sugar yield under various conditions by significantly affecting individual cane indicators (e.g., stem weight and sucrose content) and population indicators (effective stems). For achieving superior sugar yield and economic returns, the effective threshold for the application of CRF was found under D25 conditions, considering the ultimate goal of sugarcane production.

### 2.2. Response of Sugarcane Rhizosphere Soil Nutrients to Different Fertilizer Applications

The conventional nutrient content of sugarcane rhizosphere soils was significantly (*p* < 0.05) affected by different levels of CRF application. Compared with CK, the application of CRF, D1, and D15 significantly increased the soil pH and reduced the soil acidity in sugarcane succession. In the D15 treatment, the soil organic carbon reached its peak, and the sugarcane rhizosphere available nitrogen nutrients showed a corresponding trend. However, the response of both AP and AK varied depending on CRF usage. For instance, in a specific CRF application range for the field (D1–D2), the AP content of sugarcane rhizosphere soils showed a gradual decrease, indicating significant differences from the CK and D3 treatments (Table 2). Conversely, the maximum AK content corresponded to the maximum CRF application in the field.

### 2.3. Composition of Sugarcane Rhizosphere Microbial Communities and Metabolites

The composition of rhizosphere microbial communities and diversity was determined after high-throughput sequencing of sugarcane rhizosphere soil samples from different CRF applications. The results showed that the ACE index, which characterizes the alpha diversity of the bacterial community, showed significant differentiation between the D25 and D1 treatments (Appendix A), and the Chao1 index showed a significant difference between D25, CK, and D1 (Appendix A). The α-diversity of the rhizosphere fungal communities, as represented by the Shannon, ACE, and Chao1 indices (Appendix A), remained largely unchanged despite alterations in fertilizer application rates. In addition, the dilution curves of the bacterial and fungal communities, illustrated in Appendix A, indicated the complete sequencing depth, thus confirming the accuracy of the microbial community analysis. The major bacterial groups that aggregated in the sugarcane rhizosphere under different CRF application rates included Proteobacteria, Actinobacteriota, Acidobacteriota, Nitrospirota, Verrucomicrobiota, Planctomycetota, Bacteroidota, Firmicutes, and Chloroflexi. Compared with other CRF applications, the relative abundance of Proteobacteria decreased under D2 conditions, corresponding to an increase in the abundance of Actinobacteriota, Firmicutes, and a bacterial phylum that was not clearly identified. The relative abundance of Firmicutes increased gradually within a specific range of CRF use (CK-D2) and decreased gradually when applied beyond the D2 condition, as shown in Figure 1a. Furthermore, the most dominant rhizosphere fungal groups found in sugarcane were Glomeromycota, Chytridiomycota, Mortierellomycota, Rozellomycota, Basidiomycota, and Ascomycota. However, the application of CRF at concentrations above 1350 kg/ha caused a decrease in the relative abundance of the rhizosphere fungus Basidiomycota (Figure 1b). In addition, Rozellomycota demonstrated a notable increase in relative abundance under D2 conditions compared with other fertilizer application rates. Venn analysis based on the basic operational taxonomic unit (OTU) level showed that the number of bacterial core OTUs under different CRF application conditions was 1043 (Appendix A), while the number of fungal core OTUs was 243 (Appendix A). The application of CRF significantly reduced the number of inter-root bacterial OTUs in sugarcane compared with the CK treatment without CRF, which exhibited the highest number of unique OTUs. In addition, the fungal population in the CK treatment also displayed the highest number of specific OTUs. Based on the KEGG database, rhizosphere metabolites were mainly annotated to functional pathways including the biosynthesis of other secondary metabolites, carbohydrate metabolism, energy metabolism, lipid metabolism, membrane transport, metabolism of cofactors and vitamins, metabolism of terpenoids, and polyketides (Figure 1c). Annotations based on human metabolome database (HMDB) data provided additional metabolic pathways focusing on fatty acyls, carboxylic acids and derivatives, organooxygen compounds, and prenol lipids, among others (Appendix A).

### 2.4. Screening for Differential Microorganisms and Metabolites in Sugarcane Rhizosphere Soil

Non-metric multi-dimensional scaling (NMDS) and orthogonal projections to latent structure-discriminant analysis (OPLS-DA) were used to test the reproducibility of the microbiological and metabolic samples and to enable further in-depth analysis. The NMDS results showed complete separation of microbial samples under different CRF application conditions, as demonstrated in Appendix A. Similarly, the degree of separation of the samples was indicated by the principal coordinates analysis (PCoA) based on the metabolic set, where PCo1 explained 74.71%, as shown in Appendix A. To accommodate the high-dimensional nature of the metabolomic data, PLS-DA and OPLS-DA were applied to characterize the sample separation and initial screening of differential metabolites between all of the samples and selected paired comparison groups, respectively. The results showed that all treated replicate samples were discriminatory based on metabolite expression (Appendix A). Meanwhile, in the comparison groups of D25 vs. D15 and D3 vs. D1, replicate samples under the corresponding CRF application conditions were similarly significantly separated based on metabolite expression (Q2Y > 0.5). The reliability of the OPLS-DA model was demonstrated by the multiple modelling scatterplot after the permutation test (Appendix A). Under stringent LDA screening conditions, the bacterial groups *Gammaproteobacteria*, *Acidimicrobia*, *Acidibacter_ferrireducens*, *Mycoplasma*, Vibrionaceae, *Vibrio*, and *Bryobacter* were identified as being able to distinguish between different biomarkers of CRF application (Figure 2a). The fungal group showed the prominence of *Waitea*, *Enterocarpus*, *Herpotrichiellaceae*, Saccharomycetales, Debaryomycetaceae, *Candida_quercitrusa Kurtzmaniella*, and *Mycena_noctilucens* (Figure 2b).

There were variations in the quantity of significantly altered metabolites recorded for different CRF application rates (Appendix A). For instance, in D25 vs. D15, a total of 24 metabolites had significantly changed (up: 8; down: 16) as identified by screening. Comparing D3 with D1, 57 metabolites experienced significant changes (up: 24; down: 33) and were detected. Meanwhile, D2 vs. CK had the highest number of significantly altered metabolic markers of all the treatments, with 102 markers in total. Over 80% (up: 84) of the markers showed an increase in expression, while 18 showed a decrease. Three CRF application rates (high, moderate, and low) were included as treatments. Their significantly up-regulated and down-regulated metabolites were illustrated via radargrams. For example, in the comparison between D2 and D3, the following metabolites were found to have significantly increased levels in the rhizosphere soil of sugarcane: taurocholic acid, CMP-N-acetylneuraminate, Dolichotheline, GentamicinC1, Piperyline, and L-N-(1H-Indol-3-ylacetyl)asparticacid (Figure 2c). On the other hand, the metabolites with significantly lower levels in the same comparison group were Alpha-Linolenicacid, Alysifolinone, Prostaglandin F2alpha, and LolitremE, as shown in Figure 2d. Moreover, CRF application resulted in increased levels of rhizosphere chemicals such as N-Methylhistamine, Alpha-Linolenicacid, Dopaquinone, and Gingerglycolipid B, accompanied by a decrease in the levels of Rotenonic acid, 4′-Hydroxyacetophenone, myo-Inositol, Hydrogenobyrinate, and CMP-N-acetylneuraminate, compared with CK.

### 2.5. Soil Nutrients Drive the Formation of Rhizosphere Microbial Communities and Metabolite Assemblages

The redundancy analysis (RDA) at the microbial genus level and metabolites revealed the key drivers of the rhizosphere microenvironment. Conventional nutrient assays first explored possible drivers. The drivers of rhizosphere nutrients varied depending on the amount of CRF applied. In sugarcane rhizosphere bacterial compartments, soil pH, AN, and SOC were all strongly correlated with bacterial community distribution and were important for shaping rhizosphere bacterial groups under the conditions of this experiment (Figure 3a). For example, Proteobacterium and Vicinamibacteriaceae showed a positive correlation with soil pH, whereas Acidothermus, Nostocaceae, and Mycoplasma showed a negative drive with soil pH. Similarly, the sugarcane rhizosphere fungal groups were also significantly affected by soil pH, AN, and SOC, and this effect was more pronounced compared with the bacterial group (Figure 3b). *Trechispora*, *Trichoderma*, *Chaetomiaceae,* and *Tredhisporales* were all negatively correlated with soil pH, while increased AN in the rhizosphere soil promoted the enrichment of these fungal genera. The amount of CRF applied in the field contributed to soil pH, AN, and SOC as important drivers of the rhizosphere microbial community of sugarcane. In addition, for the metabolite pools, soil pH, AK, and AP were more importantly correlated with them (Figure 3b). The soil AN content showed low correlation with the identified metabolite pools. Overall, conventional rhizosphere nutrients in sugarcane were differentially driven and possibly specific for microbiota and secretion content.

### 2.6. Random Forest Screening for Characteristic Microorganisms and Metabolites under Different Fertilizer Application Conditions

To further demonstrate the impact of CRF application on sugarcane rhizosphere microcosm, the random forest algorithm was used to characterize microorganism and metabolite screening. The top 20 features were visualized based on the order of importance of the features. Among them, the rhizosphere bacterial group including *Firmicutes_bacterium*, Sphingoaurantiacus, *Buchnera*, *Cetobacterium*, *Nakamurella*, Microbacteriaceae, *Achromobacter* Oxalobacteraceae, and *Anaeromyxobacter* were accurately screened. These characterized bacteria showed great variability under different CRF application conditions. For example, *Anaeromyxobacter* had a high abundance under D1 and D25 conditions, whereas it had a lower abundance in D3 with higher CRF application and in the control without CRF (Figure 4a). Characteristic fungi screened in random forest included *Pseudaleuria*, *Kurtzmaniella*, *Epicoccum*, *Wickerhamiella*, *Erysiphe*, *Cercospora*, *Lecanicillium*, *Curvularia*, *Gymnopilus,* and *Leotiomycetes*. Similar to the bacterial compartments, there was variability in the abundance response of the characterised fungi screened for CRF application. For example, *Leotiomycetes* and *Cercospora* increased in abundance with increasing CRF application. Under D3 conditions, abundance peaked for all treatments in this study (Figure 4b). The metabolite set was similarly characterized using the algorithm. The characterized metabolites screened included isoleucyl-proline, 7-deoxyloganetate, lolitrem E, 6-methylquinoline, all-trans-heptaprenyl diphosphate, N1-34-dihydroxybenzoyl-N8-citryl-spermidine, immunomycin, glucocerebrosides, pyochelin, ginger glycolipid B, styrene, nicotinamide, avermectin A2a aglycone, and methylmalonic acid (Figure 4c). The changes in the characterized metabolites were sometimes accompanied by similar changes in the characteristic microbial genera of the sugarcane roots. For example, when comparing the CK group without CRF and the D1 experimental group with CRF application, 7-deoxyloganetate, Lolitrem E, nicotinamide, and methylmalonic acid showed a higher abundance as a result of CRF application. Similarly, *Leotiomycetes* and *Microdochium* in the fungal group and Anaeromyxobacter in the bacterial group showed an increase in abundance in response to the use of CRF.

### 2.7. Association of Dominant Microorganisms and Metabolites in Sugarcane Rhizosphere Soils

To further explore this potential simultaneous relationship, heatmaps were used to illustrate the correlation among bacteria, fungi, and metabolites. The results showed that the microbial populations of sugarcane rhizosphere soils had significant concomitant relationships. For example, the clearly identified dominant bacterial genus *Sphingomonas* showed a significant negative correlation (*p* < 0.05) with the dominant rhizosphere fungal genera (*Penicillium*, *Coprinellus*, *Aspergillus*, *Cladosporium*, *Candida,* and *Mortierella*). Reyranella showed a significant negative regulatory relationship with *Cladosporium*, *Candida*, and *Mortierella* and a significant positive regulatory relationship with *Mycena*. The dominant bacteria *Mycobacterium*, *Conexibacter*, and *Acidipila_Silvibacterium* showed a positive correlation with more fungi (Figure 5a). This relationship led to relative differences in population abundance. The regulatory role played by dominant species in the network of mutualistic relationships is, to some extent, a measure of the homeostasis of the rhizosphere microbiosphere. The role of metabolites as a bridge to rhizosphere regulation in maintaining sugarcane rhizosphere homeostasis is also irreplaceable. The heatmap results of microbial−metabolite correlations specifically show that the dominant microbial genera regulate the increase or decrease in levels of characteristic metabolites. For example, in the bacterial group, *Burkholderia*, *Bradyrhizobium,* and *Bryobacter* showed a significant negative correlation with Xanthoxic acid (Figure 5b), while in the fungal group, *Plectosphaerella* showed a significant positive correlation with Xanthoxic acid (Figure 5c). It should be clear that the numerous and complex interrelationships of sugarcane root microorganisms and metabolites do not indicate causality, but can serve as a foundation for further exploration of the molecular regulatory mechanisms.

## 3. Discussion

### 3.1. Response of Sugarcane Phenotypic Traits and Rhizosphere Conventional Soil Nutrients to CRF Applications

As observed in this study, the application rate of controlled-release fertilizer (CRF) had a significant impact on the growth of sugarcane in the field. Under different conditions, such as at D25 and D3, varying effects were observed on sugarcane plant height, stem diameter, and sucrose content. The different application rates of CRF affected individual cane indicators (e.g., stem weight and sucrose content) including population indicators such as effective stems, which highlight the trade-offs between field cane stem yield and sugar yield [27]. To optimize sugar yield and economic returns, and it is recommended to apply CRF under D25 conditions, taking into account the ultimate goal of sugarcane production (Table 1). These findings provide valuable insights for improving sugarcane cultivation practices. In the realm of agricultural research, understanding the intricate dynamics between soil properties and nutrient application is pivotal for sustainable crop production [28,29,30]. The present study delves into the nuanced impacts of controlled-release fertilizer (CRF) application on sugarcane rhizosphere soils, unraveling several intriguing dimensions that warrant further exploration. First and foremost, the profound influence of varying CRF levels on soil pH and acidity unveils the pivotal role of nutrient management in shaping soil chemistry [31]. The significant elevation in soil pH subsequent to CRF application signifies the potential of CRF to ameliorate soil conditions, fostering a favorable alkaline environment for sugarcane growth [32]. However, the intricate relationship between pH alteration and nutrient availability necessitates deeper investigation. The peak in soil organic carbon content, which is synchronized with the enhanced availability of nitrogen nutrients in the D15 treatment, raises intriguing questions about the underlying mechanisms. This may correspond to the increase in microbial activity and organic matter decomposition after CRF application [33]. The divergent responses of available phosphorus (AP) and potassium (AK) in relation to CRF utilization pose an intriguing paradox. The gradual decrease in AP content within a specific range of CRF application prompts inquiries into potential interactions between CRF and the mechanisms of soil phosphorus fixation [34]. Similarly, the direct correlation between the maximum AK content and the pinnacle of CRF application requires meticulous exploration into the intricate dynamics of potassium release, uptake, and retention (Table 2).

### 3.2. Response of Sugarcane Rhizosphere Microorganisms to Fertilizer Differences

The profound insights garnered from the high-throughput sequencing of sugarcane rhizosphere soil samples following various CRF applications offer a fascinating glimpse into the complex interplay between microbial communities and nutrient dynamics. The investigation of alpha diversity, as measured by the ACE and Chao1 indices, reveals intriguing differentiation between treatments, suggesting the potential of CRF application to modulate bacterial richness [35,36]. However, the stability of alpha diversity indices within rhizosphere fungal communities, regardless of varying fertilizer rates, sparks inquiries into the mechanisms of resilience governing fungal populations [37]. Specific fungal taxa may have unique adaptive traits in response to changes in nutrient environments [38,39]. The taxonomic composition of the rhizosphere bacterial communities further unveils the dominance of Proteobacteria, Actinobacteriota, Firmicutes, and other key groups. In the case of conventional fertilizers, rhizosphere bacteria exhibited variability regarding the application of CRF. For instance, Abdullah et al. (2022)’s sugarcane study discovered that the rhizosphere bacterial communities with varying nitrogen fertilizer application rates showed significant changes in Acidobacteria and Proteobacteria [40]. The composition of the CRF envelope and the nutrients enclosed within it could attract more functionally specific microbial populations in the root area of sugarcane, thereby resulting in varied changes at the bacterial phylum level. The enriched populations may contribute to both the breakdown and transformation of the envelope and the uptake and utilization of active nutrients [41]. Furthermore, the application of conventional chemical fertilizers results in the acidification of the soil environment, which shapes the ecological niche suitable for the survival of acidic bacteria, showing an enrichment of Acidobacteria, among others. The low enrichment of Acidobacteria in sugarcane roots under CRF implies the important role that CRF plays in alleviating soil acidification [42]. In the fungal realm, the decrease in Basidiomycota abundance and the concurrent increase in Rozellomycota under specific CRF application rates introduced intriguing questions about the ecological implications of these shifts [34,43]. Dramatic fluctuations in sugarcane rhizosphere fungal populations are often associated with plant health, and a stable rhizosphere fungal community under CRF application may indicate a trend towards healthy sugarcane growth. The integration of metabolomics data broadens this perspective, highlighting the intricate metabolic pathways affected by CRF application. Furthermore, the observed discriminatory capacity of specific microbial groups as biomarkers for CRF application underscores the potential to harness microbial signatures to gauge nutrient efficacy [44].

### 3.3. Response of Characterized Microorganisms and Metabolites Screened by Random Forests to CRF Application

The application of CRF in sugarcane rhizosphere microcosms had a significant impact on the composition of microorganisms and metabolites, as demonstrated by the analysis using the random forest algorithm. The top 20 features, including various rhizosphere bacteria, fungi, and metabolites, were accurately identified and characterized based on their importance in the system [45]. Among the characterized bacterial groups, several genera stood out for their variability in response to CRF application. For instance, *Anaeromyxobacter* exhibited a high abundance under D1 and D25 conditions, indicating a positive response to CRF. However, its abundance decreased in the presence of higher CRF application (D3) and in the control group without CRF. *Anaeromyxobacter* is known for its versatile metabolic capabilities, including the ability to degrade organic compounds and its involvement in iron cycling processes [46,47]. The observed changes in its abundance suggest its potential role in nutrient cycling and adaptation to different CRF levels. Other bacterial genera, such as *Firmicutes_bacterium*, *Sphingoaurantiacus*, *Buchnera*, *Cetobacterium*, *Nakamurella*, *Microbacteriaceae*, *Achromobacter*, and *Oxalobacteraceae*, also displayed variability in response to CRF application. Certain types of metabolites produced by *Cetobacterium* may contribute to the host organism’s ability to increase its ability to metabolize sugars and other nutrients. This phenomenon may also apply to nutrient transformation within plant tissues [48]. Alterations in the rhizosphere metabolome accompany this similar biological response process [49]. While their specific functions in the sugarcane rhizosphere are not well understood; they may contribute to nutrient cycling, plant growth promotion, or other beneficial interactions within the rhizosphere ecosystem [50]. Similar to the bacterial compartments, the characterized fungi also exhibited variability in response to CRF application [34,51]. For instance, *Leotiomycetes* and *Cercospora* showed increased abundance with increasing CRF application, reaching their highest levels under D3 conditions. *Leotiomycetes* are known to include various plant pathogens and saprophytic fungi [52], while *Cercospora* is a common foliar pathogen [53]. The increased abundance of these fungi suggests a potential shift in the fungal community structure and their potential involvement in nutrient cycling or disease dynamics under different CRF levels. However, it is unclear whether this alteration has been included in the observable span of the present study. The metabolite set characterized in this study revealed several compounds that showed changes in abundance corresponding to the CRF application. For example, isoleucyl-proline, 7-deoxyloganetate, lolitrem E, nicotinamide, and methylmalonic acid exhibited a higher abundance in response to CRF application compared with the control group without CRF. These metabolites may play important roles in nutrient uptake, defense responses, or signaling pathways in sugarcane (Figure 4). For example, methylmalonic acid plays an important role in CO_2_ fixation in photosynthesis and cellular pH regulation [54]. In plant roots, methylmalonic acid binds to calcium to form soluble calcium salts, thereby promoting calcium uptake and transport, which may correspond to the mechanism of soil-promoted calcium salt uptake in sugarcane under CRF application conditions [55].

### 3.4. Dominant Microorganisms and Metabolite Associations under CRF Application Conditions

The heatmap analysis utilized in this study has unveiled intricate correlations within the bacterial−fungal−metabolite triad, shedding light on the coexisting relationships among these key components (Figure 5). Notably, the dominant bacterial genus *Sphingomonas* exhibited a noteworthy negative correlation with the dominant rhizosphere fungal genera, namely *Penicillium*, *Coprinellus*, *Aspergillus*, *Cladosporium*, *Candida*, and *Mortierella*. This intriguing finding suggest that *Sphingomonas* and these fungal genera could be competing for similar resources or niche space, influencing their respective population abundances [56,57]. Additionally, *Reyranella* displayed distinct regulatory relationships with *Cladosporium*, *Candida*, *Mortierella*, and *Mycena*, indicating its potential role in modulating fungal dynamics within the rhizosphere [58]. Among the bacterial genera, *Mycobacterium*, *Conexibacter*, and *Acidipila_Silvibacterium* demonstrated a positive correlation with a higher abundance of fungi. This positive correlation implies a potential synergistic relationship where these bacterial taxa might provide favorable conditions for fungal growth, or vice versa [59]. The regulatory role of dominant microbial species in influencing characteristic metabolite levels underscores their significance in maintaining sugarcane rhizosphere homeostasis. For instance, *Burkholderia*, *Bradyrhizobium*, and *Bryobacter*, as representative bacterial groups, displayed a noteworthy negative correlation with Xanthoxic acid, while Plectosphaerella, within the fungal group, exhibited a positive correlation with the same metabolite. This metabolite−microbe interaction hints at potential metabolic cooperation or competition mechanisms that contribute to the intricate metabolic landscape of the rhizosphere [60]. It is essential to emphasize that the observed correlations and relationships identified in this study do not establish a direct causality. Rather, they provide a solid foundation for further exploration of the underlying molecular mechanisms driving these complex interactions. These findings beckon future research endeavors that delve into the intricate web of molecular regulatory pathways, elucidating how microbial communities and metabolites collectively shape the ecological balance within the sugarcane rhizosphere.

## 4. Materials and Methods

### 4.1. Experimental Site and Materials

The field experiment was conducted on 10 March 2021 at the Sugarcane Experimental Base of Guangxi University (N22°50′28″, E108°17′9″) using the variety “ROC22”. The specific N:P:K ratio of the CRF was 18-8-14 (N:P_2_O_5_:K_2_O), and the detailed contents of the fertilizer are given in Appendix A. The conventional fertilizer employed for control was an uncoated fertilizer with the same N, P, and K nutrients as the CRF (N:P_2_O_5_:K_2_O = 18:8:14). Guangxi is located in a monsoon climate zone with a warm climate, an average annual temperature of 20.3 °C, and abundant rainfall throughout the year. The natural climatic advantages provide sufficient water and heat conditions for sugarcane growth. The basic physical and chemical properties of the soil collected before the experiment were as follows: pH 4.45, soil organic matter (SOC) 43.10 g·kg^−1^, total nitrogen (TN) 1.24 g·kg^−1^, total phosphorus (TP) 0.55 g·kg^−1^, and total potassium (TK) 3.49 g·kg^−1^.

### 4.2. Experimental Design and Sample Collection

In order to find the appropriate field application rate of CRF for sugarcane growth, a randomized block design was used to create five groups of different application rates of film-coated CRF treatments and one group of conventional fertilizer treatments as a control. The fertilizer application rates were 675 kg/ha, 1012.5 kg/ha, 1350 kg/ha, 1687.5 kg/ha, and 2025 kg/ha, designated as D1, D15, D2, D25, and D3, respectively, and the control group, CK, was the conventional fertilizer treatment used by local farmers in Guangxi, and the fertilizer application rate was 1687.5 kg/ha. Each treatment was replicated three times in the experimental group and five times in the control group. The area of each plot was approximately 200 m^2^. Sugarcane was planted with a row spacing of 1.2 m, a plant spacing of 0.2 m, and a planting density of about 100,000 shoots/ha. During sugarcane cultivation, 40% and 60% of the total fertilizer was applied as basal (12 March 2021) and top-up fertilizer (2 July 2021). The fertilizers contained 0.1% Kill Order + 0.1% Thiamethoxam to prevent pests and diseases. A five-point sampling method was used to collect soil samples (0–20 cm). The soil samples were sealed in sterile bags and marked on the outside of the bag with the relevant treatment. After mixing the soil samples well and removing impurities, the soil samples were divided into two parts. One part was stored at −80 °C in the laboratory for the total soil DNA and metabolite extraction, and the other part was dried and stored in a naturally ventilated area for routine soil nutrient content determination [61].

### 4.3. Measurement of Phenotypic Traits and Rhizosphere Soil Physico-Chemical Indicators in Sugarcane

The number of cane plants with effective stems in the test plot was counted at the maturity stage of the cane process. A random sampling method was used to measure the height and stem diameter of sugarcane plants using a height gauge and vernier calipers. Ten randomly selected sugarcane plants within a unit area were weighed for total weight and the average weight of a single stem was calculated. The percentage of sugar in the sugarcane sample liquid was determined using a hammerometer. The formula for calculating the converted sugar content (%) from hammermetry [62]:Sucrose content (%) = hammermetry (%) × 1.0825 − 7.703

A pH meter (PB-10, Sartorius, Göttingen, Germany) was used to determine the pH of the soil suspensions (water:dry soil = 2.5:1). The externally heated potassium dichromate volumetric method was used to determine the soil organic carbon (SOC), and the alkaline dissolution diffusion method was used to measure the available nitrogen (AN) content of rhizosphere soils. The sodium bicarbonate method was used to extract soil available phosphorus (AP), and the molybdenum antimony colorimetric method was used to determine the soil content. After leaching the soil with ammonium acetate for available potassium (AK), flame photometry was used to determine the precise soil content of immediate potassium. The determination of all soil physical and chemical properties in this experiment can refer to “Soil Agrochemical Analysis” [63]

### 4.4. Soil Total DNA Extraction and High-Throughput Sequencing

The total genomic DNA was extracted from 20 samples using the TGuide S96 Magnetic Soil/Stool DNA Kit (Tiangen Biotech (Beijing) Co., Ltd., Beijing, China) according to manufacturer’s instructions. The hypervariable region V3-V4 of the bacterial 16S rRNA gene were amplified with primer pairs 338F (5′-ACTCCTACGGGAGGCAGC A-3′) and 806R (5′-GGACTACHVGGGTWTCTAAT-3′). Amplification primers targeting the fungal ITS region were ITS1F (5′-CTTGGTCATTTAGAGGAAGTAA-3′) and ITS2R (5′-GCTGCGTTCTTCATCGATGC-3′) [64]. PCR products were checked on agarose gel and purified through the Omega DNA purification kit (Omega Inc., Norcross, GA, USA). The purified PCR products were collected and the paired ends (2 × 250 bp) was performed on the Illumina Novaseq 6000 platform.

### 4.5. Quality Control and Filtering of Microbiological Data

The qualified sequences with more than 97% similarity thresholds were allocated to one operational taxonomic unit (OTU) using USEARCH (version 10.0) [65]. Taxonomy annotation of the OTUs was performed based on the Naive Bayes classifier in QIIME2 using the SILVA database [66] (release 138.1) with a confidence threshold of 70%. Alpha was performed to identify the complexity of species diversity of each sample utilizing QIIME2 (version 2020.6) software.

### 4.6. Non-Target Metabolomics Analysis

The LC/MS system for metabolomics analysis was composed of Waters Acquity I-Class PLUS ultra-high performance liquid tandem a Waters Xevo G2-XS QTof high resolution mass spectrometer. The column used was purchased from Waters Acquity UPLC HSS T3 column (1.8 μm 2.1 × 100 mm). Positive ion mode: mobile phase A: 0.1% formic acid aqueous solution; mobile phase B: 0.1% formic acid acetonitrile. Negative ion mode: mobile phase A: 0.1% formic acid aqueous solution; mobile phase B: 0.1% formic acid acetonitrile Injection volume 1 μL [67].

A Waters Xevo G2-XS QTOF high-resolution mass spectrometer was used to collect primary and secondary mass spectrometry data in MSe mode under the control of the acquisition software (MassLynx V4.2, Waters, Shanghai, China) [68]. In each data acquisition cycle, dual-channel data acquisition was performed on both low collision energy and high collision energy at the same time. The low collision energy was 2 V, the high collision energy range was 10~40 V, and the scanning frequency was 0.2 s for a mass spectrum. The parameters of the ESI ion source were as follows: Capillary voltage: 2000 V (positive ion mode) or −1500 V (negative ion mode); cone voltage: 30 V; ion source temperature: 150 °C; desolvent gas temperature 500 °C; backflush gas flow rate: 50 L/h; Desolventizing gas flow rate: 800 L/h. The raw data collected using MassLynx V4.2 is processed by Progenesis QI software (version 2.0) for peak extraction, peak alignment, and other data processing operations, based on the Progenesis QI software online METLIN database and Biomark’s self-built library for identification, and, at the same time, the theoretical fragment identification and mass deviation were all within 100 ppm.

### 4.7. Data Analysis

SPSS 26 and DPS software (version 9.01) were used to perform basic statistical analyses. Significant differences among treatments were indicated by *p* < 0.05 or other specified values of *p*. The composition of the rhizosphere bacterial community and PCoA analysis were performed using BMKCloud (www.biocloud.net (accessed on 13 August 2023)). RDA was used to visualize the relationship between microbial communities, soil environmental factors, and metabolites. The R packages ‘randomForest’ and ‘pheatmap’ were used for screening characteristic features and assessing the associations among dominant bacteria, fungi, and metabolites, respectively. The microbial differential analysis based on LDA effect size (LEfse) was performed using R (version 4.0.3). The significantly changed metabolite radargrams were generated using BMKCloud (www.biocloud.net). The identified compounds were classified and their pathways were determined using HMDB [69], KEGG [70], and lipidmaps databases [71]. Difference multiples were calculated and compared based on grouping information. The T test was used to determine the significance *p*-value of each compound. The R language package “ropls” was used to perform OPLS-DA modeling [72], and 200 permutation tests were conducted to assess the reliability of the model. The variance inflation factor (VIP) value of the model was calculated through multiple cross-validation. The method of combining the difference multiple, *p*-value, and VIP value of the OPLS-DA model was used to identify the differential metabolites.

## 5. Conclusions

Excellent yields resulting from the use of controlled release fertilizer (CRF) in sugarcane field production prompted further exploration into the molecular mechanisms of the rhizosphere microorganisms and metabolites in sugarcane under varying CRF application rates. In order to maximize sugar yield and economic returns, it is recommended to apply CRF under D25 conditions with the ultimate goal of sugarcane production in mind. Metabolite analyses showed that CRF application resulted in the use of n-methylhistamine, alpha-linolenicacid, dopaquinone, and glycolic acid in sugarcane production. Dopaquinone and gingerglycolipid B were observed as a result of sugarcane growth autoregulation and active interspecies interactions among microorganisms that affect fertilizer decomposition and utilization in the soil. The OPLS-DA modeling results confirmed the significant difference between conventional fertilizers and CRF. The use of CRF resulted in distinct metabolites and restructuring of the rhizosphere microbial community in sugarcane growth and rhizosphere soil, as revealed by the random forest analysis, distinguishing it from conventional fertilizers. *Achromobacter*, *Oxalobacteraceae*, and *Anaeromyxobacter* were particularly significant in the algorithm according to their importance. A significant inverse correlation was observed between the bacterial genus *Sphingomonas* and the primary rhizosphere fungal genera, including *Penicillium*, *Coprinellus*, *Aspergillus*, *Cladosporium*, *Candida*, and *Mortierella*. There is competition between *Sphingomonas* and fungal genera in similar ecological niches or resource spaces. After screening, future research will focus on investigating the roles of specific microbial populations and metabolite assemblages. This study contributes to our understanding of the ecological roles of sugarcane rhizosphere metabolites and rhizosphere soil microorganisms when using field-applied controlled-release fertilizers.

## Figures and Tables

**Figure 1 ijms-24-14086-f001:**
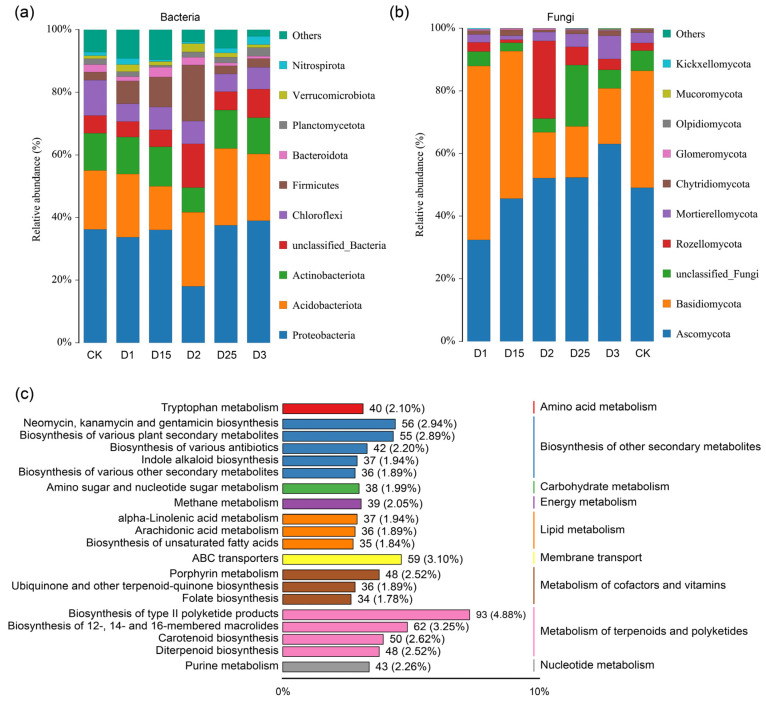
A histogram displays the relative abundance of rhizosphere bacteria (**a**) and fungi (**b**) in sugarcane under varying CRF conditions at the phylum level. The histogram depicts the distribution of the Top20 metabolic pathways of rhizosphere soil metabolites annotated according to the KEGG database (**c**), and the length of the columns corresponds to the number of metabolites annotated to the pathway. The percentage represents the proportion of metabolites within a specific pathway to the total number of annotated metabolites.

**Figure 2 ijms-24-14086-f002:**
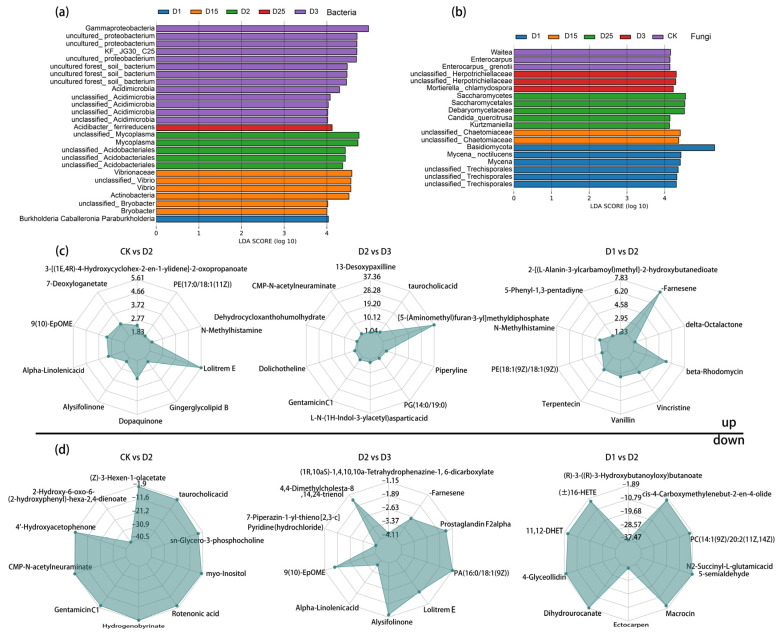
Linear discriminant analysis (LDA) based on the level of bacterial (**a**) and fungal (**b**) microbial genera showed the rhizosphere microbial classes that caused significant differences between CRF applications, LDA > 4, *p* < 0.05. Metabolite radar chart significantly up-regulated (**c**) and down-regulated (**d**) between fertilizer applications showed the classes of rhizosphere differentially secreted species, where fold change (FC) = 1, *p*-value = 0.01, Variable importance in projection (VIP) = 1, and metabolites in the top10 of each difference group were labelled. The grid lines plotted in the diagram represent log2FC. The lines that connect the log2FC values for each metabolite were shaded.

**Figure 3 ijms-24-14086-f003:**
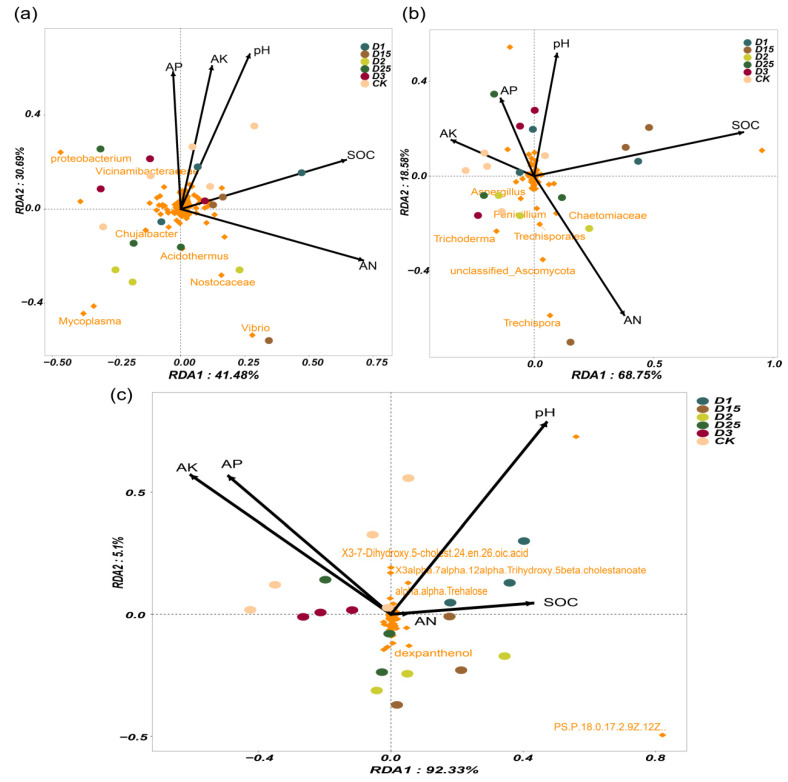
Redundancy analysis based on bacterial (**a**) and fungal (**b**) genus level and RDA based on metabolites (**c**). Different colors represent replicate samples under different CRF application conditions. Arrows represent conventional soil nutrients in the sugarcane root zone, and their lengths represent the magnitude of the correlation. The microbial relative abundance and metabolite expression abundance the top 200 were retained and are labelled in orange.

**Figure 4 ijms-24-14086-f004:**
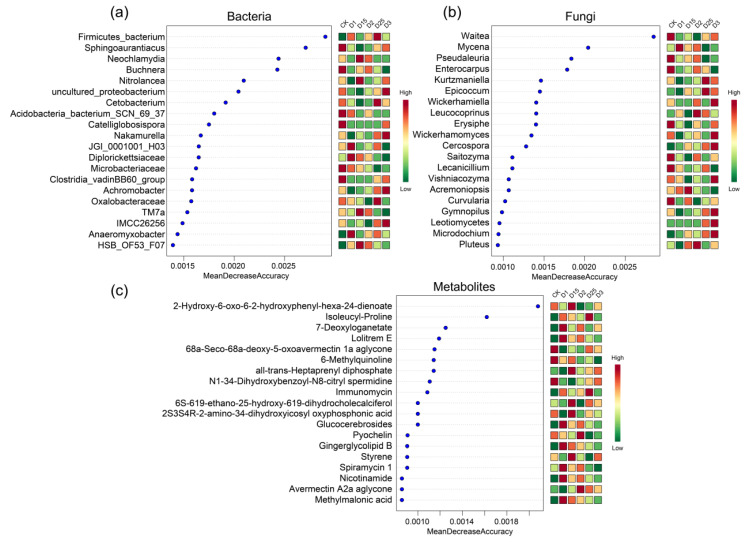
A randomized forest approach was used to screen for the bacterial genera (**a**), fungal genera (**b**), and metabolites (**c**) that contribute to the variability of different CRF applications. Horizontal coordinates are the mean importance, vertical coordinates are the microbial genera or metabolites, and heatmaps show the variability in microbial genus abundance or metabolite abundance between treatments, with the top 20 features of importance shown.

**Figure 5 ijms-24-14086-f005:**
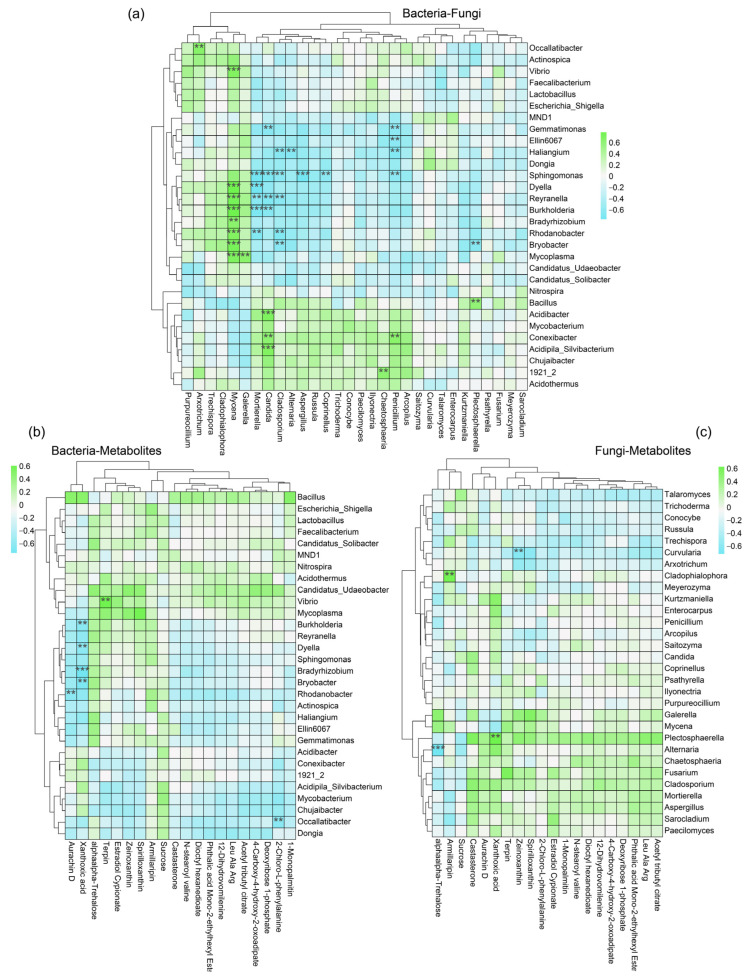
Heatmap of the correlation between two factors in the bacterial−fungal metabolite set based on Spearman’s correlation coefficient. Bacteria and fungi are the clearly identified microbial genera and belong to the top 30 with the highest degree of correlation. Similarly, the metabolites with the highest degree of correlation (top 20) were used for heatmapping. |r| > 0.6, *p* value indicates the level of significance, ** *p* < 0.01, *** *p* < 0.001. Correlations include bacteria−fungus (**a**), bacteria−metabolites (**b**) and fungi−metabolites (**c**). Blue indicates a negative correlation and green indicates a positive correlation.

**Table 1 ijms-24-14086-t001:** Effects of different application rates of CRF on the indicators of the agronomic traits of sugarcane.

Treatments	Plant Height	Stem Diameter	Sucrose Content	Stem Weight	Effective Stems	Cane Yield	Sugar Yield
	(cm)	(cm)	(%)	(kg)	(Numbers/hm^−2^)	(kg/hm^−2^)	(kg/hm^−2^)
CK	287.67 ± 17.18 b	2.68 ± 0.92 b	10.85 ± 0.64 ab	1.46 ± 0.13 bc	55,030 ± 6808 a	79,749.21 ± 6745.89 abc	8652.44 ± 893.44 bc
D1	287.92 ± 6.74 b	2.75 ± 0.24 b	10.43 ± 0.26 ab	1.54 ± 0.23 bc	40,080 ± 2324 b	68,961.30 ± 5467.62 bc	6789.10 ± 797.34 bc
D15	288.83 ± 7.75 b	2.92 ± 0.04 ab	10.50 ± 0.43 ab	1.74 ± 0.09 ab	40,950 ± 5877 b	70,961.30 ± 8467.84 bc	7441.10 ± 949.13 bc
D2	302.30 ± 5.07 ab	2.97 ± 0.14 a	10.24 ± 0.31 ab	1.90 ± 0.21 a	47,845 ± 1045 ab	90,357.16 ± 8398.16 ab	9232.12 ± 597.13 ab
D25	314.03 ± 9.74 a	3.02 ± 0.11 a	11.32 ± 0.39 a	2.04 ± 0.21 a	49,125 ± 2288 ab	100,657.59 ± 13,961.44 a	11,430.61 ± 1939.49 a
D3	304.13 ± 5.72 ab	3.07 ± 0.20 a	9.70 ± 1.13 b	2.03 ± 0.27 a	50,035 ± 9070 ab	101,594.66 ± 24,240.53 a	9827.33 ± 2404.75 ab

Note: Tabular data are presented as mean ± standard deviation. LSD test was used for test of difference. Different lower-case letters represent significant differences between treatments, *p* < 0.05.

**Table 2 ijms-24-14086-t002:** Effect of different application rates of CRF on soil nutrients.

Treatment	pH	SOC (g/kg)	AN (mg/kg)	AP (mg/kg)	AK (mg/kg)
CK	4.34 ± 0.16 b	37.93 ± 6.64 b	107.33 ± 17.96 ab	378.77 ± 136.18 a	196.36 ± 46.56 b
D1	4.97 ± 0.29 a	48.70 ± 12.75 ab	119.00 ± 37.86 ab	168.15 ± 37.70 b	170.45 ± 54.00 b
D15	4.61 ± 0.21 ab	56.06 ± 9.64 a	169.17 ± 67.26 a	159.69 ± 20.46 b	142.09 ± 25.56 b
D2	4.26 ± 0.24 b	40.19 ± 7.94 b	115.50 ± 35.00 ab	138.77 ± 5.29 b	181.01 ± 13.22 b
D25	4.37 ± 0.21 b	40.38 ± 7.08 b	101.50 ± 29.90 b	184.30 ± 38.22 b	200.99 ± 65.36 b
D3	4.38 ± 0.15 b	42.65 ± 5.14 ab	117.17 ± 1.76 ab	273.73 ± 35.86 a	302.43 ± 86.76 a

Note: Soil organic carbon, SOC; available nitrogen, AN; available phosphorus, AP; available potassium, AK. Different lowercase letters represent significant differences between treatments, *p* < 0.05.

## Data Availability

The complete data sets generated in our study have been deposited in the NCBI Sequence Read Archive database under BioProject ID: PRJNA972874.

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
