# Peer review of "Integrating the Soil Microbiota and Metabolome Reveals the Mechanism through Which Controlled Release Fertilizer Affects Sugarcane Growth"

_ijms, 2023, doi:10.3390/ijms241814086_

Round 1

Reviewer 1 Report

The manuscript "Integrating soil microbiome and metabolome reveals mechanism by which controlled release fertilizer affects sugarcane growth" by Zhaonian Yuan, Qiang Liu, Lifang Mo, Ziqin Pang and Chaohua Hu investigates the patterns of aggregation of sugarcane root-associated microorganisms and metabolite-related mechanisms during controlled release fertilizer application.

After a careful reading and judgment, I think this manuscript has some major problems that need to be reviewed:

 1. In the Abstract, the fertilizer concentrations for samples D15 and D2 are incorrect (mixed up).

2. In the Introduction, GSH - needs to be deciphered.

3. In the Introduction, the latest developments in the authors' field of research are well described, but there is no clearly formulated aim. Also, the last paragraph of the Introduction should be changed (reworded) and the authors should emphasize more strongly the novelty of their research.

4. Based on Figure 1b, the sentence "However, the application of CRF caused a decrease in the relative abundance of the rhizosphere fungus Basidiomycota (Fig. 1b)." should be replaced with " However, the application of CRF at concentrations above 1350 kg/ha caused a decrease in the relative abundance of the rhizosphere fungus Basidiomycota (Fig. 1b)." (page 4).

Or is the colour scheme in the figure mixed up?

5. In addition to the number of core OTUs, other conclusions can be drawn from the Venn diagram when analysing bacterial and fungal communities. The authors should analyse the results carefully.

6. In section 3.2 of the Discussion, the authors should pay more attention to the different response of bacterial and fungal communities to the fertilizers used (conventional fertilizers and different concentrations of CRF).

7. The use of references 38, 39, 40, 46, 47 in the Discussion should be deciphered. At present, this part of the manuscript looks more like a typical literature review than a discussion of the results obtained. In general, authors should rewrite the Discussion section.

 In addition:

1. In Figures S2 and S3, the color scheme of the samples should be the same for bacteria and fungi. For example, D1 should be blue for both bacteria and fungi.

2. Which conventional fertilizer was used as a control? Its composition should be reported.

3. Authors should check references carefully. For example, reference 61 in the Materials and Methods (primers for sequencing the ITS region) is incorrect. Reference 63 should also be changed (correct reference - Quast C, Pruesse E, Yilmaz P, Gerken J, Schweer T, Yarza P, et al. The SILVA ribosomal RNA gene database project: improved data processing and web-based tools. Nucleic Acids Res. (2013) J41:D590-596. doi: 10.1093/nar/gks1219).

4. References 66-68 should be moved to section 4.7 where the use of these databases is mentioned.

Minor editing of English language required

Author Response

Review 1

The manuscript "Integrating soil microbiome and metabolome reveals mechanism by which controlled release fertilizer affects sugarcane growth" by Zhaonian Yuan, Qiang Liu, Lifang Mo, Ziqin Pang and Chaohua Hu investigates the patterns of aggregation of sugarcane root-associated microorganisms and metabolite-related mechanisms during controlled release fertilizer application.

After a careful reading and judgment, I think this manuscript has some major problems that need to be reviewed:

  1. In the Abstract, the fertilizer concentrations for samples D15 and D2 are incorrect (mixed up).

Answer: Dear reviewer,Thank you very much for pointing out the errors in our manuscript. We sincerely apologize for such mistakes, and we have made the necessary corrections, which are highlighted in yellow. Additionally, we have reviewed the manuscript for similar issues. Once again, we appreciate you bringing our errors to our attention.

  1. In the Introduction, GSH - needs to be deciphered.

Answer: Thank you very much for your professional suggestions. As you mentioned, it is necessary to provide explanations for professional abbreviations upon their first occurrence. It was an oversight on our part, and we have made the necessary corrections while also checking for similar abbreviation issues with other professional terms in the manuscript. All the corrections have been highlighted in yellow.

  1. In the Introduction, the latest developments in the authors' field of research are well described, but there is no clearly formulated aim. Also, the last paragraph of the Introduction should be changed (reworded) and the authors should emphasize more strongly the novelty of their research.

Answer: Thank you very much for your professional advice. Your suggestions have greatly contributed to improving the quality of our manuscript. Following your recommendations, we have added a clear description of the research objectives in the "Introduction" and rephrased the final paragraph to highlight the novelty and innovation of our study. All the changes have been marked and highlighted in yellow.

  1. Based on Figure 1b, the sentence "However, the application of CRF caused a decrease in the relative abundance of the rhizosphere fungus Basidiomycota (Fig. 1b)." should be replaced with "However, the application of CRFat concentrations above 1350 kg/ha caused a decrease in the relative abundance of the rhizosphere fungus Basidiomycota (Fig. 1b)." (page 4). Or is the colour scheme in the figure mixed up?

Answer: Thank you very much for your professional advice. The color scheme of the figures is indeed correct. However, as you mentioned, the description was inaccurate regarding the decrease in relative abundance of Basidiomycota due to the extensive application of CRF. We sincerely apologize for the inappropriate wording in the manuscript, and we have made the necessary corrections to rectify this mistake. All the changes have been marked and highlighted in yellow.

  1. In addition to the number of core OTUs, other conclusions can be drawn from the Venn diagram when analysing bacterial and fungal communities. The authors should analyse the results carefully.

Answer: Thank you very much for your professional advice. It is indeed true that Venn diagrams can provide a lot of information. However, considering the extensive amount of data from the metabolome and microbiome, we have only included the most crucial details. Following your suggestions, we have further elaborated on other aspects in the description of the Venn results section to enhance the quality of the manuscript. Once again, we sincerely appreciate your professional recommendations, as they have greatly helped improve the quality of our manuscript.

  1. In section 3.2 of the Discussion, the authors should pay more attention to the different response of bacterial and fungal communities to the fertilizers used (conventional fertilizers and different concentrations of CRF).

Answer: Thank you very much for your professional advice. Following your suggestions, we have rephrased the paragraph 3.2 in the "Discussion" section. The revised description highlights the differential responses of sugarcane rhizosphere microbiota to fertilizers. All the changes have been marked and highlighted in yellow.

  1. The use of references 38, 39, 40, 46, 47 in the Discussion should be deciphered. At present, this part of the manuscript looks more like a typical literature review than a discussion of the results obtained. In general, authors should rewrite the Discussion section.

Answer: Thank you very much for your professional advice. It is true that there are issues with certain descriptions in the Discussion section, as you mentioned. In order to enhance the manuscript, we have reorganized the language in the Discussion section to discuss and explain the observed phenomena. Additionally, we have addressed the concerns regarding the references you mentioned by providing appropriate citations to support the viewpoints presented in the Discussion section.

 In addition:

  1. In Figures S2 and S3, the color scheme of the samples should be the same for bacteria and fungi. For example, D1 should be blue for both bacteria and fungi.

Answer: Thank you very much for your detailed and professional editing suggestions. Following your advice, we have made necessary corrections to the color scheme of the Venn diagram in the Supplementary Materials. Your recommendations have greatly helped us improve the quality of the manuscript.

  1. Which conventional fertilizer was used as a control? Its composition should be reported.

Answer: Thank you very much for your professional advice. The conventional fertilizer used in this study is a standard chemical fertilizer with a nutrient ratio of N:P2O5:K2O=18-8-14. The basic macronutrient composition of nitrogen, phosphorus, and potassium is consistent with controlled-release fertilizers. We have added this basic information to the "Materials and Methods" section of the manuscript. All revisions have been marked and highlighted in yellow.

  1. Authors should check references carefully. For example, reference 61 in the Materials and Methods (primers for sequencing the ITS region) is incorrect. Reference 63 should also be changed (correct reference - Quast C, Pruesse E, Yilmaz P, Gerken J, Schweer T, Yarza P, et al. The SILVA ribosomal RNA gene database project: improved data processing and web-based tools. Nucleic Acids Res. (2013) J41:D590-596. doi: 10.1093/nar/gks1219).

Answer: Thank you very much for your professional advice. We are sorry that the reference citations have caused inconvenience in your reading. We have carefully reviewed the citation format, position, and content of all references. Your constructive suggestions have greatly improved the quality of the manuscript. All changes and revisions to the references have been marked and highlighted in yellow.

  1. References 66-68 should be moved to section 4.7 where the use of these databases is mentioned.

Answer: Thank you very much for your professional advice. Following your suggestion, we have moved the three references, 66-68, to the appropriate section 4.7. As you mentioned, these databases are mentioned in section 4.7. Your recommendation has greatly helped us improve the quality of the manuscript. The modifications have been marked and highlighted in yellow.

Reviewer 2 Report

Dear Authors,

Thank you for your manuscript submission “Integrating soil microbiome and metabolome reveals mechanism by which controlled release fertilizer affects sugarcane growth.” The manuscript showed the significant potential of controlled-release fertilizer (CRF) in sugarcane field production using soil microbial community profiling and non-targeted metabolomics, which can be applied for sustainable development in the sugarcane industry. Here are my specific comments:

-       Line 14 (page 2), more references are required for the wide application of CRF

-       Some typos and grammar errors are present in the main context, such as line 12-21 (page 2) or line 30-37 (page 16), which should be revised.

-       Line 17-45 (page 2), this paragraph is confusing and should be split into different paragraphs. Each paragraph should contain only one main point

-       What is the innovation and significance of this study compared with previous studies? More details are required

-       Are there any assumptions for experimental conditions and data analysis?

-       Page 15, how did you get the equation for sucrose content? Reference is required

-       Are there any limitations for the method in this study?

-       Future research is missing in conclusion section

Some typos and grammar errors are still present in the main context and should be revised

Author Response

Review 2

Dear Authors, Thank you for your manuscript submission “Integrating soil microbiome and metabolome reveals mechanism by which controlled release fertilizer affects sugarcane growth.” The manuscript showed the significant potential of controlled-release fertilizer (CRF) in sugarcane field production using soil microbial community profiling and non-targeted metabolomics, which can be applied for sustainable development in the sugarcane industry. Here are my specific comments:

-       Line 14 (page 2), more references are required for the wide application of CRF

Answer: Thank you very much for your professional advice. As you said, more references should be provided for the wide application of controlled release fertilizers. We have re-added more references in this section.

-       Some typos and grammar errors are present in the main context, such as line 12-21 (page 2) or line 30-37 (page 16), which should be revised.

Answer: Thank you very much for your professional advice. We deeply apologize for the grammar errors found in the manuscript. In addition to addressing the specific areas you mentioned, we have also checked the grammar and spelling in other sections of the manuscript. All changes have been marked and highlighted in yellow.

-       Line 17-45 (page 2), this paragraph is confusing and should be split into different paragraphs. Each paragraph should contain only one main point

Answer: Thank you very much for your professional advice. Based on your suggestion, we have reorganized the language in the introduction section. Furthermore, due to the emphasis of this study on the interaction between microbial communities and metabolites, we have introduced the importance of both microbes and metabolites in plant growth together in the introduction section.

-       What is the innovation and significance of this study compared with previous studies? More details are required

Answer: Thank you very much for your professional advice. Based on your suggestion, we have rephrased the innovativeness and novelty of this study towards the end of the "Introduction" section. The revisions have been marked and highlighted in yellow.

-       Are there any assumptions for experimental conditions and data analysis?

Answer: Thank you very much for your professional question, which has prompted us to think more deeply. Prior to conducting this study, we had the following hypothesis: due to its unique coating structure and composition, controlled-release fertilizers may cause a significant enrichment of specialized microbial communities in the sugarcane rhizosphere. In this hypothesis, we hoped to capture compounds released by microbes or roots to aid in nutrient absorption or chemicals that facilitate coating dissolution. Additionally, we hypothesized that the long-term nutrient supply strategy of controlled-release fertilizers could promote sugarcane growth and affect sucrose accumulation. At the same time, we speculated that high doses of controlled-release fertilizers would not only fail to promote sugarcane growth, but also waste fertilizer resources and increase production costs for sugarcane farmers, reducing the economic benefits of sugarcane. The key to the widespread use of controlled-release fertilizers lies in controlling production costs, so the application rate in the experimental design is a reasonable assumed rate for us.

-       Page 15, how did you get the equation for sucrose content? Reference is required

Answer: Thank you very much for your professional advice. As you suggested, in order to ensure the scientific rigor and accuracy of the calculations, it is necessary to provide references for the sources of the methods used. We have included this information in the manuscript. Your advice has been immensely helpful in improving the quality of our manuscript.

-       Are there any limitations for the method in this study?

Answer: Thank you very much for raising targeted questions. In our microbial metagenomic sequencing, we employed a relative quantification method to quantify the sugarcane rhizosphere community. However, absolute quantification would provide a more scientifically and accurately reflective result. Additionally, non-targeted metabolomics annotation revealed a large number of metabolites. This poses challenges in observing the differences in metabolite profiles caused by experimental variations. In the future, we will gradually narrow our focus and utilize targeted metabolomics to delve deeper into the molecular mechanisms underlying the impact of controlled-release fertilizers on sugarcane growth. Furthermore, the gene expression status affecting sucrose accumulation due to long-term use of controlled-release fertilizers will also be another research direction for us.

-       Future research is missing in conclusion section

Answer: Thank you very much for your professional advice. As mentioned in our response to your previous question, deep targeted metabolomics and absolute quantification methods will be employed in future research. Additionally, validation experiments involving the supplementation of exogenous metabolites will also be an important aspect of our future studies. Your advice has prompted us to think deeply, and in accordance with your suggestion, we have added the future research directions to the final section of the "Conclusion". Once again, we appreciate your professional advice, as it has greatly contributed to the improvement of the quality of our manuscript.

Round 2

Reviewer 1 Report

The authors have taken into account all comments and observations made by the reviewer.

 Minor editing of English language required

Author Response

The authors have taken into account all comments and observations made by the reviewer.

Answer: Dear Reviewer,Thank you very much for your professional review comments. Your feedback has greatly improved the quality of our manuscript. We also appreciate your attention to our research. Your hard work serves as an excellent example for every researcher to learn from. Once again, we would like to express our gratitude for the modification suggestions you provided for our manuscript.